# Topological features without a lattice in Rashba spin-orbit coupled atoms

A. Valdés-Curiel[1], D. Trypogeorgos[1,2], Q.-Y. Liang[1], R. P. Anderson[1,3] & I. B. Spielman [1✉]

Topological order can be found in a wide range of physical systems, from crystalline solids, photonic meta-materials and even atmospheric waves to optomechanic, acoustic and atomic systems. Topological systems are a robust foundation for creating quantized channels for transporting electrical current, light, and atmospheric disturbances. These topological effects are quantified in terms of integer-valued 'invariants', such as the Chern number, applicable to the quantum Hall effect, or the $\mathbb{Z}_2$ invariant suitable for topological insulators. Here, we report the engineering of Rashba spin-orbit coupling for a cold atomic gas giving non-trivial topology, without the underlying crystalline structure that conventionally yields integer Chern numbers. We validated our procedure by spectroscopically measuring both branches of the Rashba dispersion relation which touch at a single Dirac point. We then measured the quantum geometry underlying the dispersion relation using matter-wave interferometry to implement a form of quantum state tomography, giving a Berry's phase with magnitude $\pi$. This implies that opening a gap at the Dirac point would give two dispersions (bands) each with half-integer Chern number, potentially implying new forms of topological transport.

[1] Joint Quantum Institute, University of Maryland, College Park, MD 20742, USA. [2] CNR Nanotec, Institute of Nanotechnology, 73100 Lecce, Italy. [3] La Trobe Institute of Molecular Science, La Trobe University, Bendigo, Victoria 3552, Australia. ✉email: ian.spielman@nist.gov

The topology of Bloch bands defines integers that serve to both classify crystalline materials and precisely specify properties, such as conductivity, that are independent of small changes to lattice parameters[1]. Topologically non-trivial materials first found application in metrology with the definition of the von Klitzing constant as a standard of resistance, which is now applied in the realization of the kilogram[2]. Today, topological systems have found applications in the engineering of low loss optical waveguides[3] and present a promising path to fault-tolerant quantum computation[4]. Ultracold atomic systems are an emerging platform for engineering topological lattices, from the Harper-Hofsdater model[5,6], the Haldane model[7], to the Rice-Mele model[8,9] as well as spin-orbit coupled lattices without analogues in existing materials[10,11].

A central tenet in the topological matter is the existence of integer-valued invariants that are independent of small changes to parameters. For an arbitrary closed manifold $\mathcal{M}$ and a suitable choice of a vector field (i.e., a two-form) $\mathbf{\Omega}$ the surface integral

$$\frac{1}{2\pi}\int_{\mathcal{M}}\mathbf{\Omega}\cdot d\mathbf{S} \qquad (1)$$

serves to define both the Euler characteristic and the Chern number[3,12]. When $\mathbf{\Omega}$ is equal to the local Gaussian curvature of $\mathcal{M}$, Eq. (1) yields the Euler characteristic, an invariant related to the number of handles, or genus, of $\mathcal{M}$. In contrast, when $\mathcal{M}$ is a torus describing a two-dimensional Brillouin zone (BZ) and $\mathbf{\Omega}$ is the Berry curvature characterizing the underlying quantum states, Eq. (1) instead gives the Chern number. Both the Euler characteristic and the Chern number are integer-valued, but the Euler characteristic depends only on the manifold $\mathcal{M}$ and its intrinsic curvature, whilst the Chern number depends both on a manifold (the BZ) and an additional vector field defined on $\mathcal{M}$ (the Berry curvature).

Experimental realizations of topological materials have focused on engineering different Berry curvatures in lattice systems, where $\mathcal{M}$ is always a torus due to the periodic boundary conditions of the BZ. Here we show that by eliminating the lattice potential and thereby changing $\mathcal{M}$ from $\mathbb{T}^2$ to $\mathbb{R}^2$, i.e., from a torus to a Cartesian plane, it is possible to create topological branches of the dispersion relation with half-integer generalized Chern numbers. In our experiments, we created both topological and non-topological dispersion branches by introducing Rashba-like spin-orbit coupling (SOC)[13–15] to a cold quantum gas.

We engineered Rashba SOC by resonantly coupling three internal atomic states using two-photon Raman transitions[16] as depicted in Fig. 1. The engineered system consisted of an effective spin-1/2 subspace described by a Rashba-type SOC Hamiltonian $H_{\mathrm{SOC}} = 2\alpha/m(\mathbf{q}\times\mathbf{e}_1)\cdot\hat{\boldsymbol{\sigma}}$, with added tunable higher-order terms describing quadratic and cubic Dresselhaus-like SOC[13], along with a topologically trivial high-energy branch. Here $\alpha$ is the SOC strength and $\hat{\boldsymbol{\sigma}} = (\hat{\sigma}_x, \hat{\sigma}_y, \hat{\sigma}_z)$ is the vector of Pauli operators. The total phase acquired when cyclically coupling all states adds an additional contribution to $\hat{H}_{\mathrm{SOC}}$ that is proportional to $\hat{\sigma}_z$ which can break the degeneracy of the Dirac point[13]; the tripod configuration in our experiment imprints a zero net phase to the system. Our engineered Rashba system had a single Dirac cone near $\mathbf{q} = 0$, where the two lower dispersion branches become degenerate and the Berry curvature becomes singular. Each of these branches extends to infinite momentum, making the supporting manifold a plane rather than a torus. We characterized this system using both spectroscopy and quantum state tomography. This allowed us to measure the dispersion branches and directly observe the single Dirac point linking the lowest two branches as well as to reconstruct the Berry connection to derive the associated Berry's phase.

## Results

All of our experiments started with about $10^6$ $^{87}$Rb atoms in the ground state $F = 1$ hyperfine manifold, just above the transition temperature for Bose-Einstein condensation. A bias field $B_0\mathbf{e}_3$ gave a $\omega_0/2\pi = 23.9$ MHz Larmor frequency along with a quadratic shift of $\epsilon/2\pi = 83.24$ kHz. An RF magnetic field oscillating at the Larmor frequency with strength $\Omega_{\mathrm{RF}} = 1.41(2)\epsilon$ implemented continuous dynamical decoupling (CDD)[17]. This generated a set of magnetic field insensitive states[18,19] that we denote by $|x\rangle$, $|y\rangle$ and $|z\rangle$ as they are closely related to the $XYZ$ states of quantum chemistry[20] rather than the conventional $m_F$ angular momentum states. We Raman-coupled atoms prepared in any of the $xyz$ states using the three cross-polarized 'Raman' laser beams shown in Fig. 1b, tuned to the tune-out wavelength $\lambda_L = 790$ nm, where the scalar light shift vanishes. We arranged the Raman lasers into the tripod configuration shown in Fig. 1c, bringing each pair into two-photon resonance with a single transition between the $xyz$ states with strengths $(\Omega_{zx}, \Omega_{xy}, \Omega_{yz})/2\pi = (12.6(5), 8.7(8), 10(1))$ kHz. This coupling scheme simultaneously overcomes three

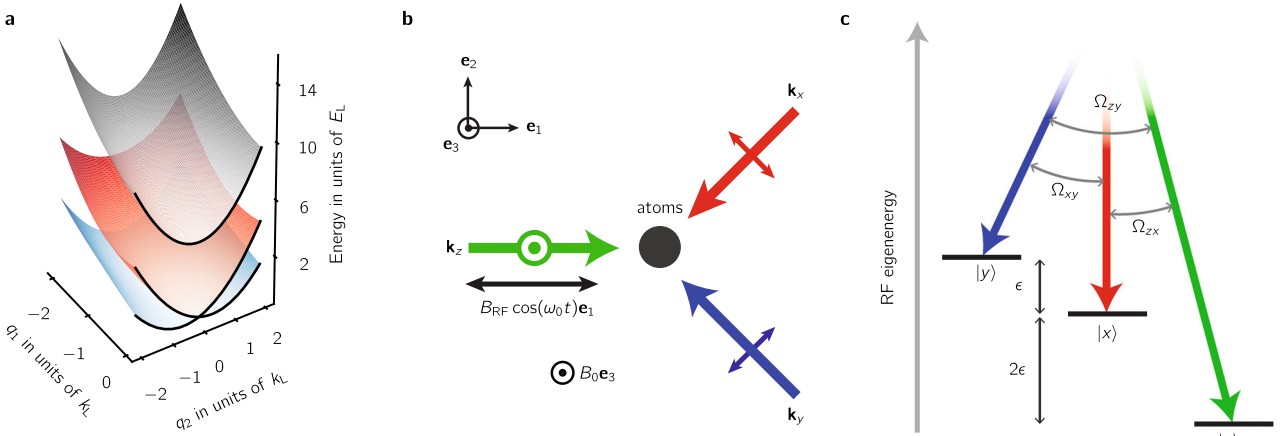

**Fig. 1 Experimental system. a** Our engineered dispersion consisted of a two-level Rashba subspace (red and blue) with a single Dirac point linking the lowest two branches and a topologically trivial higher branch (gray). **b** We generated $xyz$ states by combining a bias magnetic field $B_0\mathbf{e}_3$ with an RF magnetic field $B_{\mathrm{RF}}\cos(\omega_0 t)\mathbf{e}_1$. These states were coupled by three mutually cross-polarized Raman laser beams with wave vectors $\mathbf{k}_{x,y,z}$ and propagating along $\mathbf{e}_1$, $\mathbf{e}_2 - \mathbf{e}_1$, and $-\mathbf{e}_1 - \mathbf{e}_2$. **c** In a frame rotating with angular frequency $\omega_0$, each pair of Raman lasers was in two-photon resonance with a single transition between the $xyz$ states which we coupled strengths $(\Omega_{zx}, \Omega_{xy}, \Omega_{yz})/2\pi = (12.6(5), 8.7(8), 10(1))$ kHz.

limitations of earlier experimental realizations of two-dimensional SOC[14,15]: (1) working in the same hyperfine manifold eliminates spin-relaxation collisions[21]; (2) unlike $m_F$ states, the $xyz$ states can be tripod-coupled with lasers far detuned relative to the excited state hyperfine splitting greatly reducing spontaneous emission[20]; and (3) CDD renders the $xyz$ states nearly immune to magnetic field noise.

Each pair of Raman lasers coupled states $|i, \mathbf{k}\rangle \rightarrow |j, \mathbf{k} + \mathbf{k}_{i,j}\rangle$ where: $|i\rangle$ and $|j\rangle$ denote the initial and final $xyz$ states; $\mathbf{k}$ is the initial momentum; and $\mathbf{k}_{i,j} = \mathbf{k}_i - \mathbf{k}_j$ is the two-photon Raman recoil momentum. Dressed states with quasimomentum $\mathbf{q}$ are comprised of superpositions of three bare states $|j, \mathbf{k}\rangle$ with momentum $\mathbf{k} = \mathbf{q} - \mathbf{k}_j$. The eigenstates of our Rashba SOC Hamiltonian therefore take the form

$$|\Psi_n(\mathbf{q})\rangle = \sum_{j \in xyz} \sqrt{a_{n,j}(\mathbf{q})} e^{i\phi_{n,j}(\mathbf{q})} \Big| j, \mathbf{k} = \mathbf{q} - \mathbf{k}_j \Big\rangle, \qquad (2)$$

where the quasimomentum $\mathbf{q}$ is a good quantum number and the amplitudes are parametrized by $a_{n,j}(\mathbf{q})$ and $\phi_{n,j}(\mathbf{q})$. We leveraged the wide momentum distribution of a non-condensed ensemble ($T \approx 180$ nK and $T/T_c \approx 1.1$) to sample a wide range of momentum states simultaneously. By in addition starting separately in each of the $xyz$ states we sampled the range of quasimomentum states shown in Fig. 2a, where the momentum distributions of an initial state $|j, \mathbf{k}\rangle$ is shifted from $\mathbf{q} = 0$ by the corresponding Raman wave vector $\mathbf{k}_j$.

Our measurement protocol consisted of abruptly removing the confining potential and the Raman lasers, initiating a 21 ms time-of-flight (TOF). During this TOF we adiabatically transformed each of the $xyz$ states back to a corresponding $|m_F\rangle$ state (see "Methods") and spatially separated them using a 'Stern-Gerlach' magnetic fields gradient. Finally, we used resonant absorption imaging to measure the resulting density distributions, yielding the spin-resolved momentum distribution.

We directly measured the 2D dispersion relation using Fourier transform spectroscopy[22]. In this technique we considered the evolution of an initial state $|i, \mathbf{k}\rangle$ suddenly subjected to the Raman coupling lasers. This atomic Rabi-type interferometer is analogous to the three-port beam-splitter depicted in Fig. 2b. During a pulse time $t_p$ we followed the dynamics of the populations in the $xyz$ states evolving with oscillatory components proportional to $\sum_{j \neq n} a_{n,j}(\mathbf{q}) \cos([E_n(\mathbf{q}) - E_j(\mathbf{q})]t_p /\hbar)$, with frequencies determined by the eigenenergy differences $E_n - E_j$. Figure 2c shows the momentum-dependent populations for a fixed pulse time $t_p$ and Fig. 2d shows representative final populations as a function of $t_p$ for a fixed quasimomentum state. We Fourier transformed the populations with respect to $t_p$ and for a given quasimomentum state to produce spectral distributions as a function of quasimomentum $\mathbf{q}$. The spectral maps in Fig. 3b depict planes of constant $q_1$ in this three-dimensional distribution, whose extrema are the energy differences $E_n - E_j$ in the engineered dispersion (Fig. 1a). Together these show the presence of a single Dirac point in the Rashba subspace, evidenced by the gap closing near $\mathbf{q} = 0$ and the photon-like lower branch. The dashed curves correspond to the energy differences computed for our system using the dispersions shown in Fig. 3a, and are in clear agreement with our experiment.

However, the energies shed no light on the topology of the different branches of the dispersion, which instead requires knowledge of the eigenstates. The Berry curvature present in Eq. (1) can be derived from the Berry's connection $\mathbf{A}_n(\mathbf{q}) = i\langle \Psi_n(\mathbf{q}) | \nabla_q | \Psi_n(\mathbf{q}) \rangle$ which behaves much like a vector potential in classical electromagnetism. The Berry curvature $\mathbf{\Omega}_n(\mathbf{q}) = \nabla_q \times \mathbf{A}(\mathbf{q})$ is the associated magnetic field and the flux through any surface is the line integral of $\mathbf{A}(\mathbf{q})$ along its boundary, after neglecting the contributions of Dirac strings which we will discuss later. The Berry connection derived from Eq. (2)

$$\mathbf{A}_n(\mathbf{q}) = -\sum_{j \in \{x,y,z\}} a_{n,j}(\mathbf{q}) \nabla_q \phi_{n,j}(\mathbf{q}) \qquad (3)$$

depends on both the phase and amplitude of the wave function. We obtained $a_{n,j}(\mathbf{q})$ and $\phi_{n,j}(\mathbf{q})$ using a three-arm time-domain Ramsey interferometer, implementing a variant of quantum state tomography[23–25]. The use of a multi-path interferometer allowed us to transduce information about the relative phases into state populations, which we readily obtained from absorption images.

Figure 4a shows our experimental protocol. We adiabatically mapped an initial $|j, \mathbf{k}\rangle$ state into a corresponding eigenstate $|n, \mathbf{q} = \mathbf{k} + \mathbf{k}_j\rangle$, either in the topologically trivial highest

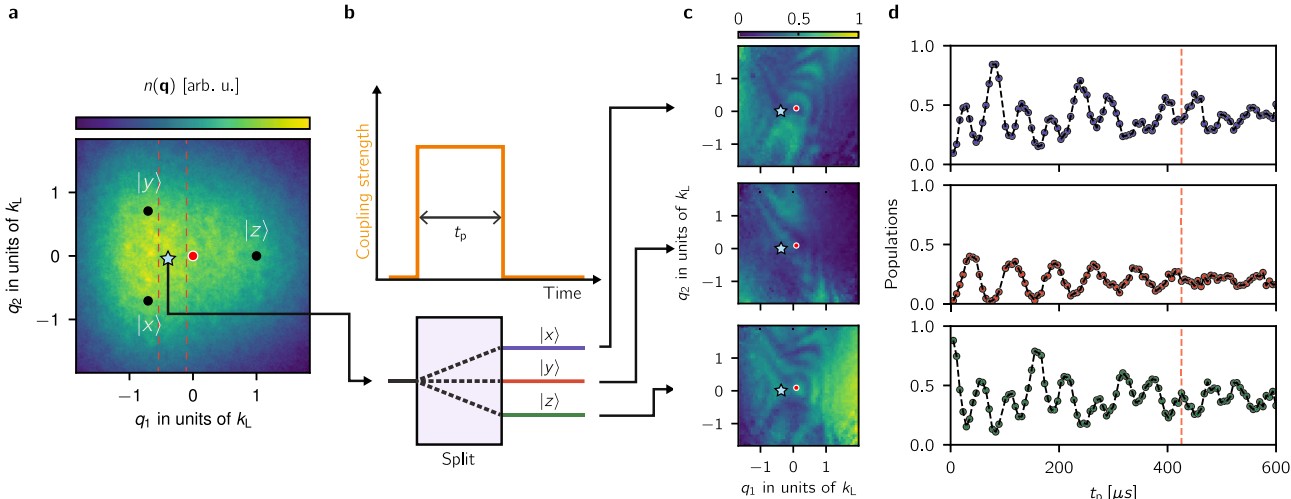

**Fig. 2 Fourier spectroscopy. a** The initial thermally occupied $xyz$ states $|j, \mathbf{k}\rangle$ lead to the displayed quasimomentum distribution. The black dots represent $\mathbf{k} = 0$ for each of the $xyz$ states which is mapped to non-zero $\mathbf{q}$, the red dot represents $\mathbf{q} = 0$ and the blue star indicates the quasimomentum $(q_1, q_2) = (-0.55, -0.18)k_L$. We used non-condensed atoms with a broad momentum distribution ($T \approx 180$ nK and $T/T_c \approx 1.1$) and performed our experiments starting separately in each of the $xyz$ states, sampling a large range of quasimomentum states. **b** We applied the Raman lasers for a variable time $t_p$: a Rabi-type atomic interferometer analogous to a three-port beam splitter. **c** Probabilities as a function of quasimomentum for a fixed Raman pulse time $t_p = 420$ μs **d** Dynamics of the final populations of the $x$ (blue), $y$ (red) and $z$ (green) states with quasimomentum $(q_1, q_2) = (-0.55, -0.18)k_L$ (red star in panels **a** and **c**) after initializing the system in the $|z\rangle$ state.

dispersion branch ($n = 3$) or in the topological ground branch ($n = 1$) by dynamically tailoring both the Raman coupling strength and detuning (see Methods). We suddenly turned off the Raman coupling, thereby allowing the three bare state components of the Rashba eigenstates to undergo free evolution for a time $t_{\text{free}}$, constituting the three arms of our time-domain interferometer. Finally we applied a three-port beam splitter using a

brief Raman 'recombination' pulse to interfere the three arms. At the end of this procedure, the population in a final state $|l, \mathbf{q}\rangle$ is

$$P_l(\mathbf{q}, t) = \sum_{i \neq j} a_{n,i} a_{n,j} \cos(\omega_{i,j}(\mathbf{q})t + \phi_{n,i}(\mathbf{q}) - \phi_{n,j}(\mathbf{q}) + \phi_{l,i,j}^{\text{p}}(\mathbf{q})),$$

(4)

which directly reads out the relative phases, independent of the output port $l$. Here $\phi_{l,i,j}^{\text{p}}(\mathbf{q})$ is a smoothly varying phase imprinted by the recombination pulse and is independent of $\mathbf{q}$ in the limit of short, strong pulses. The angular frequencies $\omega_{i,j}(\mathbf{q}) = \hbar \mathbf{q} \cdot \mathbf{k}_{i,j}/m + \delta_{i,j}$ result from the known free particle kinetic energy and detuning $\delta_{i,j}$ from the tripod resonance condition. Figure 4b shows the momentum-dependent populations in each output port at fixed $t_{\text{free}} = 160 \, \mu s$ and Fig. 4c shows the populations as a function of $t_{\text{free}}$ for a representative quasimomentum state ($q_1$, $q_2$) = (0.55, −0.92) $k_{\text{L}}$. We obtained the relative phases from Eq. (4) by fitting the measured populations to the sum of three cosines with the known free particle frequencies but unknown amplitudes and phases.

Figure 5a shows typical phase-maps for both the non-topological and topological branches. In the non-topological phase-maps the momentum dependence of the recombination pulse $\phi_{l,i,j}^{\text{p}}(\mathbf{q})$ causes a smooth variation of the phases along the Raman recoil axes that do not affect the evaluation topological index of our system. To recover the phases $\phi_{n,j}$ of the full spinor wave function from the fits, we made the gauge choice described in the "Methods" section.

We recovered the phases $\phi_{n,j}$ of the full spinor wave function from the relative phases obtained from the fits by choosing a particular gauge (see "Methods"). We then used the values of $a_{n,i}$ obtained from measuring the populations in the $xyz$ states at $t_{\text{free}} = 0$ in combination with the phases of the wave function to compute the Berry connection[26]. Figure 5b shows the three relative phases as a function of the polar angle $\phi$ for a loop of radius $q \approx 0.77 k_{\text{L}}$ for both the topological and non-topological branches. In addition to the smooth variations induced by the recombination which are present in both columns, relative phases

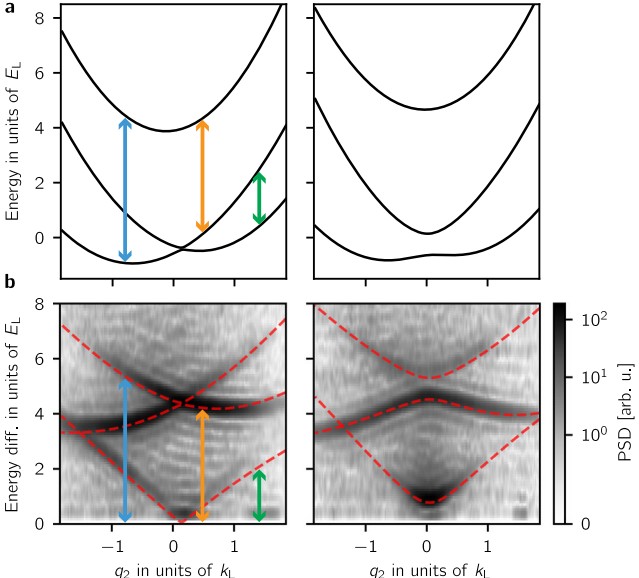

**Fig. 3 Rashba dispersion relation. a** Predicted dispersion relation as a function of $q_2$ for fixed $q_1 = -0.09 \, k_{\text{L}}$ (left) and $0.65 \, k_{\text{L}}$ (right), computed for the experiment parameters. The energy differences between the branches enclosing the vertical arrows appear as peaks in the spectral maps below. **b** Power spectral density (PSD) for the same parameters as above which we obtained by Fourier transforming the populations in the $xyz$ states with respect to $t_{\text{p}}$. The dashed lines correspond to the energy differences computed using the dispersion curves on the top panel.

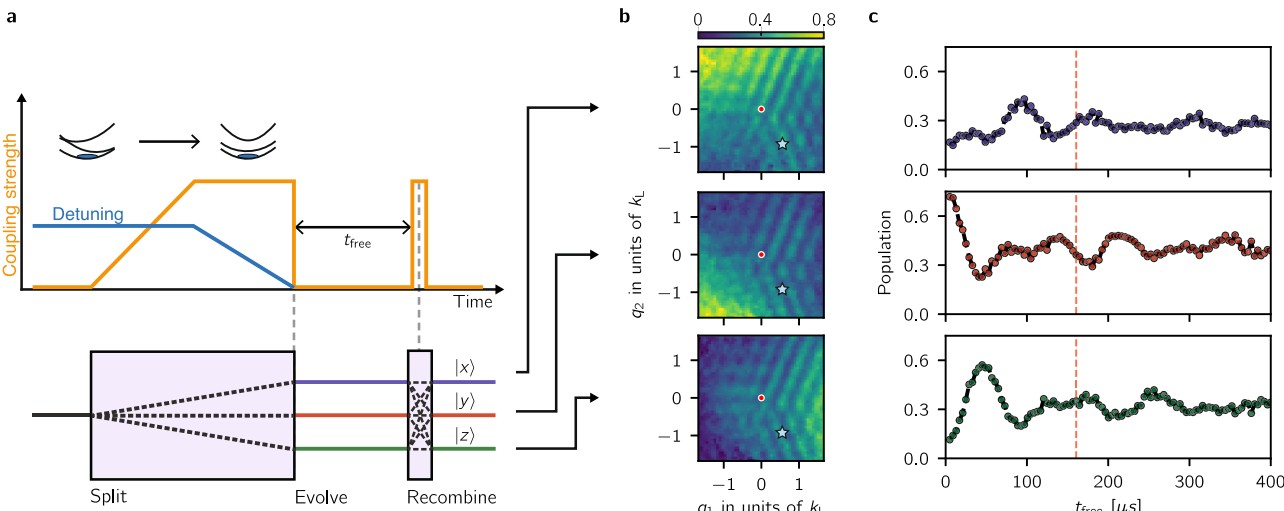

**Fig. 4 Quantum state tomography. a** Experimental protocol for three-arm Ramsey interferometer (not to scale). (Top) We started with atoms in state $|z, y, \mathbf{q}_i = \mathbf{k} + \mathbf{k}_j\rangle$ and with detuning $\delta_y = \pm 5 \, E_{\text{L}}$ and $\delta_z = \pm 5 \, E_{\text{L}}$. We ramped the Raman lasers on in $750 \, \mu s$ and then ramped the detuning to nominally zero. We let the system evolve in the dark for times between 5 and 400 μs, followed by a 25 us Raman pulse. (Bottom) The implemented experimental protocol was equivalent to a three-arm interferometer that split an initial state into three final states with amplitudes related to the initial wave function phases. **b** Population as a function of quasimomentum for the three output ports of the interferometer at $t_{\text{free}} = 160 \, \mu s$. **c** Populations in the $x$ (blue), $y$ (red) and $z$ (green) states as a function of free evolution time $t_{\text{free}}$ for an input state with quasimomentum ($q_1$, $q_2$) = (0.55, −0.92)$k_{\text{L}}$ indicated by the blue star on **b** and in the topological ground branch ($n = 1$).

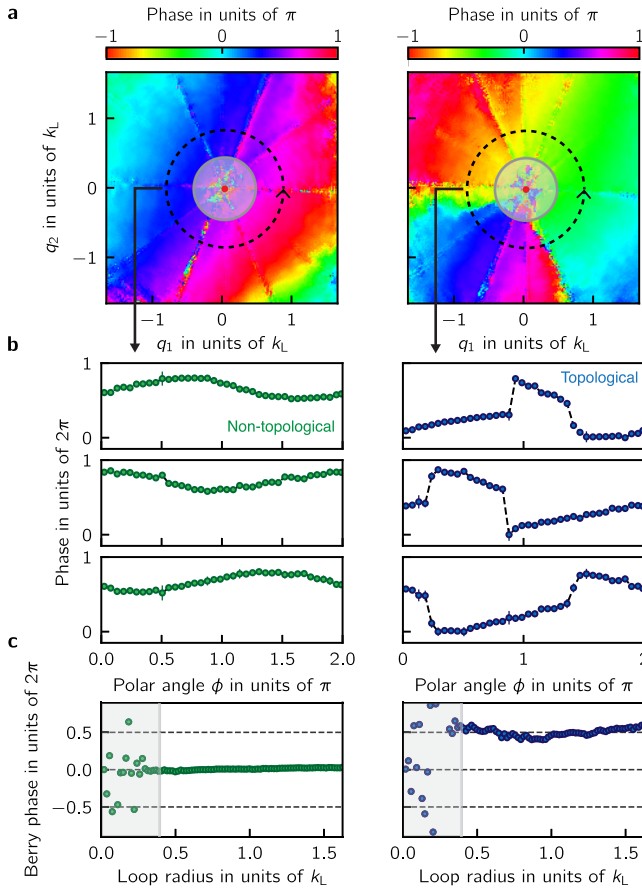

**Fig. 5 Berry phases from quantum state tomography. a** Relative phases as a function of quasimomentum from the $z \rightarrow x$ transition of the $n = 3$ non-topological branch (left) and the $n = 1$ topological branch (right). We can not reliably measure phases for $q < 0.4k_L$ due to the limited frequency resolution of the experiment. **b** Phase differences as a function of the polar angle $\phi$ for a loop radius of $0.77\,k_L$ from the $z \rightarrow x$ (top), $x \rightarrow y$ (middle) and $y \rightarrow z$ (bottom) transitions. The phases associated with the topological branch (right panels) are characterized by two $\pi$ valued discontinuities. Each row of phases was shifted by a constant value so that the three rows of phases share the same vertical axis. The error bars represent the uncertainties in the phase which were estimated using the covariance matrix of the fitting function. All phases shown here were binned and averaged using the phase uncertainties as weights. **c** Inferred Chern number as a function of loop radius. For loops with $q > 0.4k_L$ we obtained an integrated Berry phase and an inferred Chern number of $\Phi_B/2\pi = 0.01(1)$ for the non-topological branch and $\Phi_B/2\pi = 0.50(5)$ for the topological branch.

of the topological branch have two gauge independent $\pi$ valued jumps that lead to non-zero Berry phases when the Berry connection is integrated along a closed loop in momentum space. Figure 5c shows the integrated Berry phase as a function of the loop radius. The largest value of $t_{free}$ in the experiment limits how well we can resolve the phases of low frequencies $\omega_{ij}(\mathbf{q})$ near $q = 0$ as well as when two different frequencies $\omega_{ij}(\mathbf{q})$ and $\omega_{i'j'}(\mathbf{q})$ are close to each other, as can be seen in the high noise present in the phase-maps near $q = 0$ as well as in lines where the fit frequencies become nearly degenerate. This limitation is reflected in the large variation in the Berry phase depicted in the shaded region of Fig. 5c near $q = 0$. For loops with $q > 0.4\,k_L$ we obtain an integrated Berry phase $\Phi_B/2\pi = 0.01(1)$ for the non-topological branch and $\Phi_B/2\pi = 0.50(5)$ for the topological branch. However, Berry's phase measurements including ours include the

(potential) contribution of any Dirac strings traversing the integration area. In our system, these are possible at the Dirac point (red dot near $q = 0$) and contribute $\pm 2\pi$ to $\Phi_B$. Even with this $2\pi$ ambiguity, we can associate a half-integer Chern number with the topological branch. While the Chern number is only defined for gapped systems, the local Berry connection associated with the dispersion relation of our Rashba system becomes indistinguishable from that of a gapped Dirac point for large enough quasimomenta; the Dirac point can be gapped by adding additional lasers to the coupling scheme or changing the laser polarization[15]. Unlike previous measurements of $\pi$ valued Berry's phases in crystalline materials[27,28], our measurements do not rely on the adiabatic motion of particles within a band. A total Berry's phase $\Phi_B/2\pi = 0.50(5)$ integrated over the whole BZ is possible only for a topological dispersion branch in the continuum.

**Discussion.** In conventional lattices—for example, graphene, or the topological Haldane model—it is well established that Dirac points each contribute a Berry's phase of $\Phi_B/2\pi = \pm 1/2$[27], but crystalline materials conspire for these to appear in pairs[29], always delivering integer Chern numbers. In contrast, our continuum system contains a single Dirac point, resulting in a non-integer Chern number. This leads to intriguing questions about edge states at interfaces with non-integer Chern numbers with non-integer Chern number differences. Initial studies in the context of electromagnetic waveguides[30] and atmospheric waves[31] have applied Chern invariants and the bulk-edge correspondence to continuous media.

While the true Rashba Hamiltonian features a ring of degenerate eigenstates, our implementation including the quadratic and cubic Dresselhaus-like SOC that lifts this macroscopic degeneracy giving three nearly degenerate minima[13]. Already these three minima could allow the study of rich ground-state physics in many-body systems of bosons, for example, the formation of fragmented BECs[32] when the system does not condense into a single-particle state. Furthermore, the use of additional spin states or larger Raman couplings can partially restore this degeneracy allowing the possible realization of fractional Hall-like states[33]. Finally, by introducing quenches in the location of the Rashba subspace our system can be used to study other topological indices such as the linking number[34–36].

## Methods

**System preparation.** Our experiments began with $N \approx 1 \times 10^6$ $^{87}$Rb atoms in a crossed optical dipole trap[37], with frequencies $(f_1, f_2, f_3) \approx (70, 85, 254)$ Hz. We initially prepared the atoms in the $|F = 1, m_F = -1\rangle$ state of the $5S_{1/2}$ electronic ground state. We then transferred the atoms either to $m_F = 0$ or $m_F = +1$ by applying an RF field with ~20 kHz coupling strength and ramping a bias magnetic along $\mathbf{e}_3$ from 36 µT lower value to $B_i = 3.39(9)$ mT in 50 ms. We prepare the $xyz$ states by starting in each of the $m_F$ states in a bias field 72 µT lower than $B_0$ and then ramping on the RF dressing field to $\Omega_{RF}/2\pi = 117(2)$ kHz in 1 ms and then ramped the bias field to its final value $B_0 = 3.40(9)$ mT in 3 ms. We finally waited for 40 ms for the fields to stabilize prior to applying any Raman coupling. We adiabatically converted the $xyz$ states back to the $m_F$ states in TOF before imaging. We first ramped the bias field back to $B_i$ in 2 ms and then turned off the RF field in 1 ms. Finally we apply the Stern-Gerlach gradient to spatially separate the $|m_F\rangle$ states for 14 ms.

**Raman coupling the $xyz$ states.** The Raman coupling originates of the $xyz$ states, like in the regular $m_F$, from the vector light shift from the local Raman field on the ground hyperfine manifold, which can be cast into an effective magnetic Hamiltonian

$$\hat{H}_{eff} = \frac{g_F \mu_B}{\hbar} \mathbf{B}_{eff} \cdot \hat{\mathbf{F}}, \qquad (5)$$

with

$$\mathbf{B}_{eff} = \frac{iu_v}{g_s \mu_B}(\mathbf{E}^* \times \mathbf{E}) \qquad (6)$$

where $g_s$ is the spin gyromagnetic, ratio, $g_F$ the Landé g-factor, $\mu_B$ the Bohr

magneton and $u_v$ the vector polarizability

$$u_v = \frac{|\langle l ||\mathbf{d}|| l' \rangle|^2}{12}\left(\frac{1}{\Delta_{3/2}} - \frac{1}{\Delta_{1/2}}\right). \tag{7}$$

Here $\Delta_{3/2}$ and $\Delta_{1/2}$ represent the laser detuning from the $P_{3/2}$ and $P_{1/2}$ electronic states and $\langle l ||\mathbf{d}|| \rangle = \langle l = 0|\mathbf{d}|l' = 1\rangle$ is the reduced dipole matrix element.

In th frame rotating at the RF dressing frequency $\omega_0$ and after applying a rotating wave approximation, the Raman matrix elements are $\Omega_{ij} = \langle i|\hat{H}_{\text{eff}}|j\rangle = |\Omega_{ij}|e^{i\phi_{ij}}$. If $\mathbf{E}^* \times \mathbf{E}$ has non-zero projections along $\mathbf{e}_1$, $\mathbf{e}_2$ and $\mathbf{e}_3$ it is possible to drive transitions between all of the $xyz$ states. In contrast, the $m_f$ states can only be coupled when $\mathbf{E}^* \times \mathbf{E}$ lies on a plane perpendicular to the quantization axis and therefore the $xyz$ states are better suited to use different laser geometries producing dispersion relations that are closer to the true Rashba ring-like dispersion.

The energies of the $xyz$ states are $\omega_x = 0$ and $\omega_{z,y} = -(\epsilon \pm \sqrt{4\Omega_{\text{RF}}^2 + \epsilon^2})/2$. We set the frequencies of the Raman lasers to $\omega_x = \omega_L + \omega_0 + \omega_{xy}$, $\omega_y = \omega_L + \omega_0$ and $\omega_z = \omega_L - \omega_{zx}$ such that all transitions between the $xyz$ states were in two-photon resonance in the RF rotating frame. Here $\omega_L = 2\pi c/\lambda_L$ and $(\omega_{zx}, \omega_{xy}, \omega_{zy})/2\pi = (166.47, 83.24, 249.71)$ kHz are the transition frequencies between pairs of dressed states are integer multiples of $\epsilon$ for our coupling strength $\Omega = \sqrt{2}\epsilon$.

The Raman-coupled states are well described by the combined kinetic and light-matter Hamiltonian

$$\hat{H}(\mathbf{q}) = \sum_{i\in\{xyz\}}\left(\frac{\hbar^2(\mathbf{q} - \mathbf{k}_i)^2}{2m} + \hbar\delta_i\right)|i\rangle\langle i| + \sum_{i\neq j}\hbar\Omega_{ij}|j\rangle\langle i|, \tag{8}$$

where $\mathbf{k}_i$ are the Raman wave vectors, $\delta_i$ is a detuning from Raman resonance and $\Omega_{ij}$ is the Raman coupling strength between a pair of RF dressed states. For a detailed derivation of the Rashba Hamiltonian see ref. [16].

The location of the Rashba subspace is determined by the phase sum $\bar{\phi} = (\phi_{zx} + \phi_{xy} + \phi_{yz})/3$ [16]. If $\bar{\phi} = 0$ the Dirac point connects the $n = 1$ and $n = 2$ branches, if $\bar{\phi} = \pi$ the Dirac point connects the $n = 2$ and $n = 3$ and for all other values of $\bar{\phi}$ the Dirac point becomes massive (non-degenerate). For the tripod coupling scheme where each laser drives two transitions, the contribution of individual laser phases is canceled and $\bar{\phi}$ is constrained to take the values 0 or $\pi$ depending on the magnitude of $|\omega_x - \omega_y|$ and $|\omega_y - \omega_z|$ relative to the RF frequency $\omega_0$ and the sign of the vector polarizability $u_v$ [16]. In all of our experiments $\bar{\phi} = 0$. The Raman phases could be used to continuously tune $\bar{\phi}$ by adding additional Raman lasers to our setup such that the individual laser phases are not canceled out.

Our implementation of Rashba SOC is implemented entirely within the ground hyperfine manifold and has the advantages of reduced losses from spin-relaxation collisions and increased stability against environmental fluctuations due to the clock-like nature of the $xyz$ sates. The measured spontaneous emission limited lifetime of our system is 320(17) ms. The lifetime is reduced to 40(2) ms when the Raman couplings are resonant, which we attribute to technical noise in the relative phase between the RF dressing field and the Raman laser fields. We did not phase lock our Raman lasers.

The duration of all experiments with the $xyz$ states was limited to 50 ms due to overheating of the antenna producing the RF fields. We did not observe any decay from the $x$ or $y$ states into the ground $z$ during this time.

**Floquet effects**. We operated in a regime where the transition energies between the $xyz$ states were integer multiples of $\omega_{xy}$: $\omega_{zx} = 2\omega_{xy}$ and $\omega_{zy} = 3\omega_{xy}$, and used Floquet theory for a complete description of our system[38]. The Hamiltonian in Eq. (8) is therefore an effective Hamiltonian that describes the stroboscopic dynamics of the full Floquet Hamiltonian. We observed that the effective Raman coupling strengths for the driven three-level system differed from our calibrations which were performed by only driving one pair of states because of the presence of nearby quasi-energy manifolds. This effect would be mitigated for larger values of $\omega_{xy}$ as the spacing between quasi-energy manifolds is increased.

**Combining spectral maps from different states**. In the Fourier spectroscopy experiments, we initialized the system in any of the three $xyz$ states. We individually computed the Fourier transforms with respect to $t_p$ for a total of nine distributions of $|j, \mathbf{q}\rangle$ states (accounting for each of the three $xyz$ states that were split each into three states). We computed the spectral maps displayed in Fig. 2b by averaging the PSD of each distribution, where each $\mathbf{q}$ state was weighted by the mean population in $t_p$.

**State preparation for Ramsey interferometer**. For the Rashba dressed states preparation we started with RF dressed states with a different coupling strength $\Omega_{\text{RF}}/\pi 2 \pm 20$ kHz. This change shifted the energies of the $|z\rangle$ and $|y\rangle$ states by about $\pm 18.8$ kHz. The change in the $xyz$ state eigenenergies corresponded to non-zero $\delta_z$ and $\delta_y$ in Eq. (8). We chose the detuning such that the initial state had a large overlap with either the $n = 1$ or the $n = 3$ eigenstates of Eq. (8). We ramped the Raman on in 750 $\mu$s and then ramped $\Omega_{\text{RF}}$ to its final value in 1 ms, effectively ramping $\delta_z$ and $\delta_y$ close to zero. This method allowed us to prepare dressed states

in either the $n = 1$ or $n = 3$ by initializing the system in the $|y\rangle$ or $|z\rangle$ states. When we prepared the system in $|x\rangle$ the final dressed state corresponded to the $n = 2$ branch.

**Adiabaticity in state preparation**. The state preparation was not adiabatic in the vicinity of the Dirac point. The detuning ramp in the state preparation protocol had the additional effect of moving the location Dirac point through the atoms, thereby creating a detuning dependent trajectory where the state preparation was not adiabatic. We combined data where the ground state preparation had different initial states and a different detuning values (different Dirac point trajectories). Near the final location of the Dirac point ($\mathbf{q} = 0$) the state preparation can not be adiabatic regardless of the initial state or detuning used for the ground state preparation.

**Combining phases from different states**. The phases of the fitted populations at the output of the interferometer correspond to $\Delta\phi_{n,i,j,l} = \phi_{n,i}(\mathbf{q}) - \phi_{n,j}(\mathbf{q}) + \phi^P_{l,i,j}(\mathbf{q})$. The last term in the expression has $\mathbf{q}$-independent term that depends on the final state and a $\mathbf{q}$-dependent term that has no dependence on the final state, i.e., $\phi^P_{l,i,j}(\mathbf{q}) = \phi^{P_0}_l + \phi^{P_1}_{i,j}(\mathbf{q})$. When combining the phases from different initial states we removed their final state dependence by shifting $\Delta\phi_{n,i,j,l}$ by a constant number such that they maximally overlap, effectively making $\phi^{P_0}_l$ the same for all states. Finally, we averaged all the phase differences obtained from the fits, weighted by the inverse of the uncertainties obtained from the fitting procedure. For the topological branch data, we excluded the regions away from $\mathbf{q} = 0$ where the Dirac point was moved from the average. Finally we chose a gauge such that $\phi_1(\mathbf{q}) = 0$ and used this to convert phase differences into phases.

## Data availability
The datasets generated during and/or analyzed during the current study are available from the corresponding author on reasonable request.

## Code availability
The code used for analysis during the current study is available from the corresponding author on reasonable request.

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

## Acknowledgements
We appreciated conversations with W.D. Phillips, M. Fleischhauer, T. Ozawa, and J. E. H. Braz. This work was partially supported by the AFOSRs Quantum Matter MURI, NIST, and the NSF through the PFC at the JQI.

## Author contributions
I.B.S., D.T., and A.V.C. proposed the research direction. A.V.C. led the data taking and analysis effort. A.V.C., D.T., Q.Y.L., and R.P.A. contributed to the smooth operation of the experimental apparatus. All authors contributed to the witting of the manuscript and participated in useful discussions.

## Competing interests
The authors declare no competing interests.
