## [Peer Review File · Nature Communications]

Reviewer #1 (Remarks to the Author):

The manuscript "Unconventional topology with a Rashba spin-orbit coupled quantum gas" by Valdes-Curiel et al., describes the measurement of a Berry's phase of π in the setting of ultracold quantum gases, which indicates a half-integer value for the topological invariant. Half-integer Chern numbers are not allowed in lattice geometries as one cannot capture half of a monopole in the closed manifolds of the torus of the BZ. The authors overcome this by isolating a single Dirac cone, which is naturally associated to a π -Berry phase, contributing 'half' of the Chern number of the band, in the absence of the periodic potential.

In particular, they engineer a Rashba-type SOC Hamiltonian by Raman coupling a three-level (xyz) system which itself consists of dynamically-coupled hyperfine levels. This additional layer of xyz-state engineering provides them with the flexibility and coherence required for the Berry phase measurement described in the manuscript. Since SOC has been essentially already realized in cold atoms, this experimental capability of reduced spontaneous emission or immunity to magnetic fields counts toward the 'new physics' that the manuscript might lead to. The authors develop a modified state tomography technique similar to the one implemented in lattice systems, for the measurement of the engineered dispersion relations and topology as captured by their eigenstates.

I enjoyed reading this work; the experiment is nice, the analysis comprehensive, together with a relatively well-explained presentation. Although this is not the first experimental realization of a SOC in cold atoms, the current study follows a different implementation and puts forward the direct measurement of the Berry phase encoded in the eigenstates with remarkable precision. Isolating a single Dirac cone opens up the possibility of studying the unique aspects of topology outside of the torus geometry of the BZ. The work under consideration implements quenching protocols for the measurement of the wavefunction. Making sudden changes in the Hamiltonian and following the time evolution of the state has not only proved to be a powerful tool for the measurement of the underlying topology of optical lattice systems in recent years, but also brought new topological invariants and connections into light. I find the experiment an important step forward for the study of topologically protected aspects of matter in cold atoms and the results are of high interest to the wider audience suitable for Nature Communications. I have a few comments which I would like to see addressed before making a final recommendation toward publication.

-I appreciate the authors' sensitivity in calling the energy levels dispersion branches throughout the text, however I find the title of the manuscript misleading. Although the experimental technique and their measurements are significant, there is nothing unconventional in having a π -Berry phase associated with a single Dirac cone. Band touching is at a single Dirac point as engineered by the SOC, the associated Berry phase corresponds to a half-integer value.

-Since the SOC Hamiltonian corresponds to a pseudospin-1/2, would it be possible to picture the dynamics of the system to some level on the Bloch sphere for simplicity?

-The manuscript might benefit from a contrast with the studies in lattice counterparts, in particular with the quench dynamics and state tomography. They might want to cite the theory proposal for tomography [Phys. Rev. Lett. 113, 045303 (2014)] along with its experimental implementation [ref.29]. Although it is true that the experiments in lattice systems generically focused on engineering Berry curvatures, quenching techniques opened up new possibilities where the invariant is a Chern-Simons invariant defined in a higher torus formed with the addition of the time circle [Phys. Rev. Lett.121,250403 (2018); Nat. Commun.10, 1728 (2019); arXiv:1904.11656]. Can they comment on where the continuum system stands with respect to the lattice counterparts?

-They claim to be adiabatically mapping an initial state to an eigenstate for the measurement of the Berry phase. I do not understand how they are doing it in the topologically non-trivial branch where there is a Dirac point. Is the minimum radius required in Fig.5 for the reliable phase measurement also a result of the non-adiabaticity?

Reviewer #2 (Remarks to the Author):

In the manuscript by Valdes-Curiel et al., the authors implemented two-dimensional spin-orbit coupling for ultracold bosons by combining a few pairs of Raman transition beams and dynamics coupling scheme (by oscillating Bfield, I.e. energy level). They experimentally showed the existence of single Dirac point through band spectroscopy (they called it Fourier spectroscopy), and measured the topology of the energy band near the Dirac point through the quantum state tomography. A series of the measurements reported in this work validated their claim that the realized topological system have a half-integer Chern number. Generally speaking, the paper is well-written providing various instructive discussions for readers. All figures were carefully chosen, and the data sets seem high quality.

However, I have a big concern that the authors may oversell their result. The authors mentioned their central claim that they demonstrated the topological system with a half-integer Chern number by removing the lattice geometry on page 3. This may mislead the reader to think, " This work demonstrates 2D spin-orbit coupling in bulk for the first time." My concern might be due to the fact that the authors did not clearly state what new features can be achieved in their implementation compared to the previous works in Ref.[21,22]. In terms of "implementation", I don't see a big conceptual advancement compared to the previous 2D experiment (e.g Nature physics 2016 or Ref[21]). On the technical side, several minor improvements have been taken such as XYZ states and dynamic decoupling. [note (1) the (similar) 2D spin-orbit coupled system was already realized with ultracold fermions [Ref[21]] (2) the quantum state tomography was demonstrated by Hamburg group in a 2D lattice system.]

Related to above-mentioned concerns, I do see that the most interesting aspect of the work is to measure the Berry phase using quantum state tomography. It comes to my attention that the authors try to highlight their result by emphasising the direct "measurement" of the topology in the bulk 2D spin-orbit-coupled system, saying "Unconventional topology with a" in the title. To me, however, the overall findings reported in this work seem rather incremental than substantial (note that most of techniques used in this work are well-established ones). In this regard, the author may consider to revise the title to "Direct measurement of unconventional topology...." , which precisely describes what they have achieved. If the authors want to distinguish their work in regard to unconventional topology, it is highly recommended to "demonstrate" the physical consequence (e.g. transport) arising from the unconventional topology but it might be difficult to do it at this moment. In summary, I cannot recommend the publication in the current form.

A few minor questions:

- It would be greatly helpful if the authors give a bit more information about the way how the authors manage their oscillating Bfields and the Raman fields. To me, they should be phase locked otherwise it will "efficiently" heat up the system in contrast to the bare Raman system. It is mentioned in the supplementary note regarding this issue, but (maybe I missed..) it is not clear to me if the relative phase was locked or not. If yes, does is

- In the perspective, the authors mentioned that it is conceivable to check an interesting topological transport in the future. How good/bad is the heating effect if one adiabatically load ultracold bosons into the spin-orbit coupled system reported in tis work? If I understood correctly, the XYZ state must be insensitive to environment noises like a clock transition. Does Floquet term heat up the system? If not, how favourable is the parameter window in this regard?

Reviewer #3 (Remarks to the Author):

In this paper, the authors report the realization of two-dimensional Rashba spin-orbit coupling in cold atomic gas and measure the energy band and Berry phase using time-domain Ramsey interferometer. Although two-dimensional Rashba spin-orbit coupling has been realized with the similar scheme for three tripod energy levels coupling with three Raman lasers, this work presents

an improved protocol by means of RF dressed state, which has several advantages. This paper is well written and the presentation is clear. Since Rashba spin-orbit coupling is a hot topic and has broad interests, I think that this work is suitable for NC.

Here, I still want to give some technical comments and hope it is helpful for this work.

1) This work only uses thermal gas (above the transition temperature for BEC) to study the energy band and Berry phase and never use the feature of quantum gas. Suggest the title does not use "quantum gas".

2) The xyz states are coupled by three cross-polarized Raman laser beams as shown in Fig. 1b. Question A: It seems that the polarization of Raman laser k_x and k_y are parallel to paper plane in Fig. 1b? "cross-polarized" means that the polarization of Raman laser k_x and k_y have an angle between them? Question B: Does it still require the orthogonal polarization of Raman lasers for the transition of $\Delta m_F=1$ as the normal bare Zeeman states (Raman transition rule)? Here, the xyz states are dressed state. Does it still need the orthogonal polarization of Raman lasers in here? In other words, the parallel polarization of two Raman lasers still can drive the transition in this scheme. The author should give the discussion.

3) The Dirac point locating in the lower or higher two bands depends to the sign of three Raman coupled strength. In Ref. 21, it is pointed that Dirac point locates in the lower two energy bands when the frequency of Raman laser is between D1 and D2 lines, and in the higher two bands when blue detuning of D2 line and red detuning of D1 line. Does the scheme in this work still follow this rule? Authors should add the discussion about the location of Dirac point.

4) Authors point this scheme simultaneously overcomes three limitations of earlier experiments. However, everything has double sides. Does this scheme has some disadvantages?

5) As we all know, the excited states in the dressed state picture are not stable and easily decay to ground state due to collision or other mechanism. This work should give the lifetime when atoms are populated in excited state x or y .

6) "During this TOF we adiabatically transformed each of the xyz states back to a corresponding $|m_F\rangle$ state and spatially separated them using Stern-Gerlach magnetic fields gradient." Authors give the exact time of adiabatically transformed process and Stern-Gerlach in here.

7) Fig. 3: Fig. 3b gives three energy band, however, only two vertical arrows show the energy differences in Fig. 3a. In order to make clear, third vertical arrow should be added in Fig. 3a.

8) Fig. 5: The horizontal axis of Fig. 5(a) is labeled with q_2 . I think it is a writing error.

September 15, 2020

Dear Referees,

We appreciate your clearly careful reading of our manuscript, the constructive commentary, and the interest in the topic presented.

Before turning to the specific responses, there were general concerns regarding the title of the manuscript. In response to these comments, we changed the title to “Rashba spin-orbit coupled atoms: topological features without a lattice”.

Specific responses:

Reviewer 1

The manuscript “Unconventional topology with a Rashba spin-orbit coupled quantum gas” by Valdes-Curiel et al., describes the measurement of a Berry’s phase of π in the setting of ultracold quantum gases, which indicates a half-integer value for the topological invariant. Half-integer Chern numbers are not allowed in lattice geometries as one cannot capture half of a monopole in the closed manifolds of the torus of the BZ. The authors overcome this by isolating a single Dirac cone, which is naturally associated to a π -Berry phase, contributing ‘half’ of the Chern number of the band, in the absence of the periodic potential.

In particular, they engineer a Rashba-type SOC Hamiltonian by Raman coupling a three-level (xyz) system which itself consists of dynamically-coupled hyperfine levels. This additional layer of xyz-state engineering provides them with the flexibility and coherence required for the Berry phase measurement described in the manuscript. Since SOC has been essentially already realized in cold atoms, this experimental capability of reduced spontaneous emission or immunity to magnetic fields counts toward the ‘new physics’ that the manuscript might lead to. The authors develop a modified state tomography technique similar to the one implemented in lattice systems, for the measurement of the engineered dispersion relations and topology as captured by their eigenstates.

I enjoyed reading this work; the experiment is nice, the analysis comprehensive, together with a relatively well-explained presentation. Although this is not the first experimental realization of a SOC in cold atoms, the current study follows a different implementation and puts forward the direct measurement of the Berry phase encoded in the eigenstates with remarkable precision. Isolating a single Dirac cone opens up the possibility of studying the unique aspects of topology outside of the torus geometry of the BZ. The work under consideration implements quenching protocols for the measurement of the wavefunction. Making sudden changes in the Hamiltonian and following the time evolution of the state has not only proved to be a powerful tool for the measurement of the underlying topology of optical lattice systems in recent years, but also brought new topological invariants and connections into light. I find the experiment an important step forward for the study of topologically protected aspects of matter in cold atoms and the results are of high interest to the wider audience suitable for Nature Communications. I have a few comments which I would like to see addressed before making a final recommendation toward publication.

-I appreciate the authors’ sensitivity in calling the energy levels dispersion branches throughout the text, however I find the title of the manuscript misleading. Although the experimental technique and their measurements are significant, there is nothing unconventional in having a π -Berry

phase associated with a single Dirac cone. Band touching is at a single Dirac point as engineered by the SOC, the associated Berry phase corresponds to a half-integer value.

Authors: We agree that a single Dirac point and its associated π valued Berry phase contributes a half-integer to the total Chern number. What is unconventional about our system is that there is a *single* Dirac point, so that the Berry phase for all loops is either zero or π . When the single Dirac point is gapped the integrated Berry curvature is just $\frac{1}{2}$. We removed the word unconventional from the title and the whole of the document.

-Since the SOC Hamiltonian corresponds to a pseudospin-1/2, would it be possible to picture the dynamics of the system to some level on the Bloch sphere for simplicity?

Authors: It is possible to picture the dynamics of pseudospins within the Rashba subspace on the Bloch sphere. Our measurements however involved dynamics within the three dispersion branches and therefore we should look at the full wave functions.

-The manuscript might benefit from a contrast with the studies in lattice counterparts, in particular with the quench dynamics and state tomography. They might want to cite the theory proposal for tomography [Phys. Rev. Lett. 113, 045303 (2014)] along with its experimental implementation [ref.29]. Although it is true that the experiments in lattice systems generically focused on engineering Berry curvatures, quenching techniques opened up new possibilities where the invariant is a Chern-Simons invariant defined in a higher torus formed with the addition of the time circle [Phys. Rev. Lett. 121, 250403 (2018); Nat. Commun. 10, 1728 (2019); arXiv:1904.11656]. Can they comment on where the continuum system stands with respect to the lattice counterparts?

Authors: We added the reference for the tomography theory proposal [Phys. Rev. Lett. 113, 045303 (2014)]. It would be possible to perform similar experiments to those reported in *Phys. Rev. Lett.* 121, 250403 (2018); *Nat. Commun.* 10, 1728 (2019); *arXiv:1904.11656* by quenching the Raman phases sum that determines the location of the Rashba subspace. We now say this in the outlook of the paper and extended the “Raman coupling the xyz states” sub-section in the Methods to explain the effect of the laser phases on the location of the Dirac point.

-They claim to be adiabatically mapping an initial state to an eigenstate for the measurement of the Berry phase. I do not understand how they are doing it in the topologically non-trivial branch where there is a Dirac point. Is the minimum radius required in Fig.5 for the reliable phase measurement also a result of the non-adiabaticity?

Authors: The referee is correct in pointing out that it is not possible to adiabatically map an initial state into the ground state where there is a Dirac point and the minimum radius in Fig.5 is partly a result of this non adiabaticity (the other reason is that the duration of the free-evolution time limits the frequency resolution of our measurement). We added a footnote in the main text that explicitly mentions this point. We additionally added a new subsection “Adiabaticity in state preparation” to the Methods that addresses this issue in more detail.

Reviewer 2

In the manuscript by Valdes-Curiel et al., the authors implemented two-dimensional spin-orbit coupling for ultracold bosons by combining a few pairs of Raman transition beams and dynamics coupling scheme (by oscillating Bfield, I.e. energy level). They experimentally showed the existence of single Dirac point through band spectroscopy (they called it Fourier spectroscopy), and measured the topology of the energy band near the Dirac point through the quantum state

tomography. A series of the measurements reported in this work validated their claim that the realized topological system have a half-integer Chern number. Generally speaking, the paper is well-written providing various instructive discussions for readers. All figures were carefully chosen, and the data sets seem high quality.

However, I have a big concern that the authors may oversell their result. The authors mentioned their central claim that they demonstrated the topological system with a half-integer Chern number by removing the lattice geometry on page 3. This may mislead the reader to think, “This work demonstrates 2D spin-orbit coupling in bulk for the first time.” My concern might be due to the fact that the authors did not clearly state what new features can be achieved in their implementation compared to the previous works in Ref.[21,22]. In terms of “implementation”, I don’t see a big conceptual advancement compared to the previous 2D experiment (e.g Nature physics 2016 or Ref[21]). On the technical side, several minor improvements have been taken such as XYZ states and dynamic decoupling. [note (1) the (similar) 2D spin-orbit coupled system was already realized with ultracold fermions [Ref[21]] (2) the quantum state tomography was demonstrated by Hamburg group in a 2D lattice system.]

Related to above-mentioned concerns, I do see that the most interesting aspect of the work is to measure the Berry phase using quantum state tomography. It comes to my attention that the authors try to highlight their result by emphasising the direct “measurement” of the topology in the bulk 2D spin-orbit-coupled system, saying “Unconventional topology with a” in the title. To me, however, the overall findings reported in this work seem rather incremental than substantial (note that most of techniques used in this work are well-established ones). In this regard, the authour may consider to revise the title to “Direct measurement of unconventional topology....”, which precisely describes what they have achieved. If the authors want to distinguish their work in regard to unconventional topology, it is highly recommended to “demonstrate” the physical consequence (e.g. transport) arising from the unconventional topology but it might be difficult to do it at this moment. In summary, I cannot recommend the publication in the current form.

Authors: We do not claim this is the first experimental realization of 2D spin-orbit coupling and we have modified the text and now explicitly say “earlier realizations of 2D spin-orbit coupling” when citing the works in Refs [21,22]. We explicitly mention the advantages of our coupling scheme in the manuscript: (1) working in the same hyperfine manifold eliminates spin-relaxation collisions; (2) unlike mF states, the xyz states can be tripod-coupled with lasers far detuned relative to the excited state hyperfine splitting greatly reducing spontaneous emission; and (3) CDD renders the xyz states nearly immune to magnetic field noise. The first two points are the most important as they increase the lifetime of the system which is important to observe meaningful many-body physics. The Raman matrix coupling elements for mf states within the same hyperfine manifold become vanishingly small at large Raman detuning (this is necessary to keep spontaneous emission low) and without the CDD states the only option to implement a tripod coupling with far detuned lasers is to work using states within different hyperfine manifolds. The fantastic experiments in Ref. [21] used atoms in the $F=9/2$ and $F=7/2$ hyperfine manifolds of ^{40}K and such spin mixtures are susceptible to two-body collisional decay from the $7/2$ to $9/2$ ground hyperfine state (spin-relaxation collisions, see new ref [28]) which reduces the lifetime of the system. Using our implementation, it is possible to use detuned lasers to couple states within the same hyperfine manifold and as a result it is possible to have longer lived 2D spin-orbit coupled systems as is necessary to observe the many-body interaction and transport related phenomena mentioned in the outlook. We added an extended discussion on “Raman coupling the xyz states” explaining this in more detail in the Methods. Finally, we changed the title so that is better suited to describe our results.

A few minor questions:

- It would be greatly helpful if the authors give a bit more information about the way how the authors manage their oscillating B fields and the Raman fields. To me, they should be phase locked otherwise it will “efficiently” heat up the system in contrast to the bare Raman system. It is mentioned in the supplementary note regarding this issue, but (maybe I missed..) it is not clear to me if the relative phase was locked or not. If yes, does is

Authors: The Raman fields were not phase locked and we now explicitly mention it in the “Raman coupling the xyz states” subsection in the Methods. We also expanded this subsection to address the effect of the Raman field phases on the dispersion relation.

- In the perspective, the authors mentioned that it is conceivable to check an interesting topological transport in the future. How good/bad is the heating effect if one adiabatically loads ultracold bosons into the spin-orbit coupled system reported in this work? If I understood correctly, the XYZ state must be insensitive to environment noises like a clock transition. Does Floquet term heat up the system? If not, how favourable is the parameter window in this regard?

Authors: We observed a reduced lifetime compared to the spontaneous emission limited lifetime of the bare atomic states. The reduced lifetime including spontaneous emission (and possible Floquet effects) is 40 ms. Because the ~100 kHz Floquet frequency is quite high compared to the other energy scale, it is unlikely to cause heating (for example a pair of lattice laser beams detuned by 100 kHz create a uniform time-averaged potential and no observable heating, and we have used a dithered laser beam at 100 kHz to create a time-averaged potential with unchanged lifetime). The overall wavefunction phases of this system are sensitive to the relative phase between the Raman beams, as well as the RF phase. Similar to changing a uniform Peierls phase in a lattice leaves the overall dispersion unchanged, but creates effective electric forces, here changing phases leave dispersion relation unchanged but give rise to generalized forces. Because we did not stabilize these phases, their time-dependance over 10’s of ms certainly lead to heating.

Reviewer 3

In this paper, the authors report the realization of two-dimensional Rashba spin-orbit coupling in cold atomic gas and measure the energy band and Berry phase using time-domain Ramsey interferometer. Although two-dimensional Rashba spin-orbit coupling has been realized with the similar scheme for three tripod energy levels coupling with three Raman lasers, this work presents an improved protocol by means of RF dressed state, which has several advantages. This paper is well written and the presentation is clear. Since Rashba spin-orbit coupling is a hot topic and has broad interests, I think that this work is suitable for NC.

Here, I still want to give some technical comments and hope it is helpful for this work.

1) This work only uses thermal gas (above the transition temperature for BEC) to study the energy band and Berry phase and never use the feature of quantum gas. Suggest the title does not use “quantum gas”.

Authors: We changed “quantum gas” to “atoms” in the title.

2) The xyz states are coupled by three cross-polarized Raman laser beams as shown in Fig. 1b. Question A: It seems that the polarization of Raman laser k_x and k_y are parallel to paper plane in

Fig. 1b? “cross-polarized” means that the polarization of Raman laser k_x and k_y have an angle between them? Question B: Does it still require the orthogonal polarization of Raman lasers for the transition of $\Delta m_F=1$ as the normal bare Zeeman states (Raman transition rule)? Here, the xyz states are dressed state. Does it still need the orthogonal polarization of Raman lasers in here? In other words, the parallel polarization of two Raman lasers still can drive the transition in this scheme. The author should give the discussion.

Authors: By cross-polarized we mean that all three polarization vectors are orthogonal. The orthogonal polarization is necessary, just as in the m_F states, because of the $\mathbf{E} \times \mathbf{E}^*$ dependence of the vector light shift that gives rise to the Raman coupling. Unlike for the m_F states, a pair of Raman beams whose cross product is parallel to the bias field can still drive a Raman transition. The complete derivation of the dressed Raman coupling can be found in ref. 23 (D L Campbell and I B Spielman 2016 New J. Phys. 18 033035) in the main text. We expanded the “Raman coupling the xyz states” subsection of the Methods to describe this in more detail.

3) The Dirac point locating in the lower or higher two bands depends to the sign of three Raman coupled strength. In Ref. 21, it is pointed that Dirac point locates in the lower two energy bands when the frequency of Raman laser is between D1 and D2 lines, and in the higher two bands when blue detuning of D2 line and red detuning of D1 line. Does the scheme in this work still follow this rule? Authors should add the discussion about the location of Dirac point.

Authors: The location of the Dirac point is determined by the sum of the phases in the Raman couplings ij . This phase depends on the frequency of the Raman laser with respect to the D1 and D2 lines (like in Ref 21) and in the sign of the Relative frequencies with respect to the RF frequency 0. In our implementation this phase is equal to zero leading to the Dirac point being located at the lower two bands. We now explain this in more detail in the “Raman coupling the xyz states” subsection of the Methods.

4) Authors point this scheme simultaneously overcomes three limitations of earlier experiments. However, everything has double sides. Does this scheme has some disadvantages?

Authors: The main disadvantages are: (1) gradients in RF from imperfect coils can lead to decoherence, (2) the RF coils tend to overheat limiting the duration of experiments, depending on the application phase locking might be required; and (3) The two of the Raman fields involve a second rotating frame transformation that decreases their strength in the XYZ basis [detailed in *Rashba realization: Raman with RF*; D. L. Campbell and I. B. Spielman; New Journal of Physics **18** 33035 (2016)].

5) As we all know, the excited states in the dressed state picture are not stable and easily decay to ground state due to collision or other mechanism. This work should give the lifetime when atoms are populated in excited state x or y .

Authors: We limit the duration of our experiments with dressed atoms to 50 ms (much higher than the duration of any experiment reported here) as the coils producing the high RF fields tend to overheat. During this time, we did not observe any decay from atoms populated in the excited x or y states. We now mention this in the “Raman coupling the xyz states” subsection of the Methods.

6) “During this TOF we adiabatically transformed each of the xyz states back to a corresponding $|m_F\rangle$ state and spatially separated them using Stern-Gerlach magnetic fields gradient.” Authors give the exact time of adiabatically transformed process and Stern-Gerlach in here.

Authors: We expanded the “System preparation” section of the Methods to include the times for the adiabatic transformation of xyz states back to $|m_F\rangle$ as well as the duration of the Stern-Gerlach pulse.

7) Fig. 3: Fig. 3b gives three energy band, however, only two vertical arrows show the energy differences in Fig. 3a. In order to make clear, third vertical arrow should be added in Fig. 3a.

Authors: We added a third vertical arrow so that all the energy differences in Fig 3 are clearly indicated.

8) Fig. 5: The horizontal axis of Fig. 5(a) is labeled with q_2 . I think it is a writing error.

Authors: We corrected this error.

Reviewer #1 (Remarks to the Author):

I have read the updated manuscript and the authors' reply to the comments raised by the referees during the first round of the review. Overall, I find that the changes implemented by the authors have contributed to the accessibility of the manuscript and I am satisfied with their replies. The updated title addresses the overselling concerns and now directly corresponds to a description of their experiment. I believe the new subsections they included are well fitting in explaining and highlighting some experimental details. I find that the comments of Referee 2 stimulated the authors in articulating their main findings in a way better emphasising the differences with previous SOC systems, as well as the technical details given in response to Referee 3 helpful. So, I recommend publication in Nature Communications.

Reviewer #2 (Remarks to the Author):

In their revised manuscript, the authors have improved several aspects of their paper, in particular the title that is better suited to deliver the main claim. I think the authors addressed most of my concerns satisfactorily, and also other concerns raised by other Referees. The remaining question is whether the work warrants publication in Nature Comm., in light of journal's aim to publish research that is exceptional and/or of importance.

The authors have strongly argued that their manuscript represents the new approach to the synthetic spin-orbit coupling in bulk. I now agree with that statement in the revised manuscript. In this sense, it is reasonable to consider this paper an important milestone. I am therefore inclined to recommend publication.

Reviewer #3 (Remarks to the Author):

Authors have made the improvement and answered the questions in this version. However, I still have several critical questions.

- 1) In the Supplementary materials, "We set the frequencies of the Raman laser to $\omega_x = \omega_L + \omega_0 + \omega_{xy}$, $\omega_y = \omega_L + \omega_0$ and $\omega_z = \omega_L - \omega_{zx}$ ". These frequencies of three Raman lasers made me confusion, which is inconsistent with Fig. 1c. From Fig. 1c, the frequencies of three Raman lasers should be $\omega_x = \omega_L$, $\omega_y = \omega_L - \omega_{xy}$ and $\omega_z = \omega_L + \omega_{xz}$ in the dressed pictures of xyz states. I do not understand why ω_0 is added in the ω_x and ω_y , not in ω_z .
- 2) Following point 1, the sentence " $\overline{\phi}$ depends only on the detuning of the Raman relative to the RF frequency ω_0 and the sign of the vector polarizability". What's the meaning of the detuning of the Raman relative to the RF frequency ω_0 ? Is it two-photon detuning?
- 3) "We could continuously tune $\overline{\phi}$ by adding additional lasers to our setup." Please explain in detail how to tune $\overline{\phi}$ continuously by adding additional lasers.

November 3, 2020

Dear Referees,

We appreciate your clearly careful reading of our manuscript, the constructive commentary, and the interest in the topic presented. The referees indicated that their concerns have been addressed and that the manuscript would be a good match for *Nature Communications*. We have addressed the remaining questions raised by Referee #3 which we believe have helped to improve the clarity of our manuscript.

Specific responses:

Reviewer #1 (Remarks to the Author):

I have read the updated manuscript and the authors' reply to the comments raised by the referees during the first round of the review. Overall, I find that the changes implemented by the authors have contributed to the accessibility of the manuscript and I am satisfied with their replies. The updated title addresses the overselling concerns and now directly corresponds to a description of their experiment. I believe the new subsections they included are well fitting in explaining and highlighting some experimental details. I find that the comments of Referee 2 stimulated the authors in articulating their main findings in a way better emphasising the differences with previous SOC systems, as well as the technical details given in response to Referee 3 helpful. So, I recommend publication in Nature Communications.

Reviewer #2 (Remarks to the Author):

In their revised manuscript, the authors have improved several aspects of their paper, in particular the title that is better suited to deliver the main claim. I think the authors addressed most of my concerns satisfactorily, and also other concerns raised by other Referees. The remaining question is whether the work warrants publication in Nature Comm., in light of journal's aim to publish research that is exceptional and/or of importance. The authors have strongly argued that their manuscript represents the new approach to the synthetic spin-orbit coupling in bulk. I now agree with that statement in the revised manuscript. In this sense, it is reasonable to consider this paper an important milestone. I am therefore inclined to recommend publication.

Reviewer #3 (Remarks to the Author):

Authors have made the improvement and answered the questions in this version. However, I still have several critical questions.

1) In the Supplementary materials, "We set the frequencies of the Raman laser to $\omega_x = \omega_L + \omega_0 + \omega_{xy}$, $\omega_y = \omega_L + \omega_0$ and $\omega_z = \omega_L - \omega_{xz}$ ". These frequencies of three Raman lasers made me confusion, which is inconsistent with Fig. 1c. From Fig. 1c, the frequencies of three Raman lasers should be $\omega_x = \omega_L$, $\omega_y = \omega_L - \omega_{xy}$ and $\omega_z = \omega_L + \omega_{xz}$ in the dressed pictures of xyz states. I do not understand why ω_0 is added in the ω_x and ω_y , not in ω_z .

Authors: The frequencies reported in the Supplementary materials were the ones used in the laboratory while frequencies and energies in Figure 1 are represented in a rotating frame where the Hamiltonian was transformed using the unitary operator $U = \exp(-i\omega_0 F_z)$. Any term in the Hamiltonian that is proportional to F_x or F_y is transformed and gets its frequency shifted by ω_0 while the terms proportional to F_z remain unchanged. We did not add ω_0 to ω_z so that the Raman interaction was resonant after applying the unitary transformation. We modified the caption of Figure 1 to reflect the fact that these energies are represented in a rotating frame and clarified this point in the supplementary materials as well.

2) Following point 1, the sentence “ $\overline{\phi}$ depends only on the detuning of the Raman relative to the RF frequency ω_0 and the sign of the vector polarizability”. What’s the meaning of the detuning of the Raman relative to the RF frequency ω_0 ? Is it two-photon detuning?

Authors: We modified the text to clarify this point. The “detuning” here corresponds to how the magnitude of $|\omega_x - \omega_y|$ and $|\omega_y - \omega_z|$ compare to the the RF frequency ω_0 . We now also include a reference to [D L Campbell and I B Spielman 2016 New J. Phys. 18 033035] where all the steps in the derivation the Rashba Hamiltonian are explained in detail.

3) “We could continuously tune $\overline{\phi}$ by adding additional lasers to our setup.” Please explain in detail how to tune $\overline{\phi}$ continuously by adding additional lasers.

Authors: We clarified this point in the supplementary material. In our current setup $\overline{\phi}$ is independent of the individual laser phases because in the tripod coupling scheme each laser drives two transitions, resulting in the laser phase contributions cancelling out. By adding more lasers to the setup such that at least one laser is not involved in more than one transition is possible to use the laser phase to tune $\overline{\phi}$.

Reviewer #3 (Remarks to the Author):

I now agree with that statement in the revised manuscript. I am therefore recommend publication.